# Unbiased Teacher for Semi-Supervised Object Detection

**Yen-Cheng Liu**[1,2]*, **Chih-Yao Ma**[2], **Zijian He**[2], **Chia-Wen Kuo**[1], **Kan Chen**[2],
**Peizhao Zhang**[2], **Bichen Wu**[2], **Zsolt Kira**[1], **Peter Vajda**[2]
[1]Georgia Tech, [2]Facebook Inc.
{ycliu,cwkuo,zkira}@gatech.edu,
{cyma,zijian,kanchen18,stzpz,wbc,vajdap}@fb.com

## Abstract

Semi-supervised learning, *i.e.*, training networks with both labeled and unlabeled data, has made significant progress recently. However, existing works have primarily focused on image classification tasks and neglected object detection which requires more annotation effort. In this work, we revisit the Semi-Supervised Object Detection (SS-OD) and identify the pseudo-labeling bias issue in SS-OD. To address this, we introduce ***Unbiased Teacher***[1], a simple yet effective approach that jointly trains a student and a gradually progressing teacher in a mutually-beneficial manner. Together with a class-balance loss to downweight overly confident pseudo-labels, Unbiased Teacher consistently improved state-of-the-art methods by significant margins on *COCO-standard*, *COCO-additional*, and *VOC* datasets. Specifically, Unbiased Teacher achieves $6.8$ absolute mAP improvements against state-of-the-art method when using $1\%$ of labeled data on MS-COCO, achieves around 10 mAP improvements against the supervised baseline when using only $0.5, 1, 2\%$ of labeled data on MS-COCO.

## 1 Introduction

The availability of large-scale datasets and computational resources has allowed deep neural networks to achieve strong performance on a wide variety of tasks. However, training these networks requires a large number of labeled examples that are expensive to annotate and acquire. As an alternative, Semi-Supervised Learning (SSL) methods have received growing attention (Sohn et al., 2020a; Berthelot et al., 2020; 2019; Laine & Aila, 2017; Tarvainen & Valpola, 2017; Sajjadi et al., 2016; Lee, 2013; Grandvalet & Bengio, 2005). Yet, these advances have primarily focused on image classification, rather than object detection where bounding box annotations require more effort.

In this work, we revisit object detection under the SSL setting (Figure 1): an object detector is trained with a single dataset where only a small amount of labeled bounding boxes and a large amount of unlabeled data are provided, or an object detector is jointly trained with a large labeled dataset as well as a large external unlabeled dataset. A straightforward way to address Semi-Supervised Object Detection (SS-OD) is to adapt from existing advanced semi-supervised image classification methods (Sohn et al., 2020a). Unfortunately, object detection has some unique characteristics that interact poorly with such methods. For example, **the nature of class-imbalance in object detection tasks impedes the usage of pseudo-labeling.** In object detection, there exists foreground-background imbalance and foreground classes imbalance (see Section 3.3). These imbalances make models trained in SSL settings prone to generate biased predictions. Pseudo-labeling methods, one of the most successful SSL methods in image classification (Lee, 2013; Sohn et al., 2020a), may thus be biased towards dominant and overly confident classes (background) while ignoring minor and less confident classes (foreground). As a result, adding biased pseudo-labels into the semi-supervised training aggravates the class-imbalance issue and introduces severe overfitting. As shown in Figure 2, taking a two-stage object detector as an example, **there exists heavy overfitting on the fore-**

---

*Work done partially while interning at Facebook.
[1]Code: https://github.com/facebookresearch/unbiased-teacher.

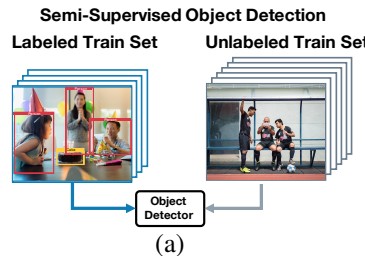
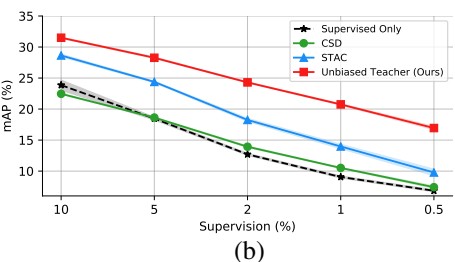

|   |   |
|:-:|:-:|
| (a) | (b) |

Figure 1: (a) Illustration of semi-supervised object detection, where the model observes a set of labeled data and a set of unlabeled data in the training stage. (b) Our proposed model can efficiently leverage the unlabeled data and perform favorably against the existing semi-supervised object detection works, including CSD (Jeong et al., 2019) and STAC (Sohn et al., 2020b).

**ground/background classification in the RPN and multi-class classification in the ROIhead** (but not on bounding box regression).

To overcome these issues, we propose a general framework – ***Unbiased Teacher***: an approach that jointly trains a *Student* and a slowly progressing *Teacher* in a mutually-beneficial manner, in which the Teacher generates pseudo-labels to train the Student, and the Student gradually updates the Teacher via Exponential Moving Average (EMA)[2], while the Teacher and Student are given different augmented input images (see Figure 3). Inside this framework, (i) we utilize the pseudo-labels as explicit supervision for both RPN and ROIhead and thus alleviate the overfitting issues in both RPN and ROIhead. (ii) We also prevent detrimental effects due to noisy pseudo-labels by exploiting the Teacher-Student dual models (see further discussion and analysis in Section 4.2). (iii) With the use of EMA training and the Focal loss (Lin et al., 2017b), we can address the pseudo-labeling bias problem caused by class-imbalance and thus improve the quality of pseudo-labels. As the result, our object detector achieves significant performance improvements.

We benchmark Unbiased Teacher with SSL setting using the MS-COCO and PASCAL VOC datasets, namely *COCO-standard*, *COCO-additional*, and *VOC*. When using only $1\%$ labeled data from MS-COCO (*COCO-standard*), Unbiased Teacher achieves $6.8$ absolute mAP improvement against the state-of-the-art method, STAC (Sohn et al., 2020b). Unbiased Teacher consistently achieves around 10 absolute mAP improvements when using only $0.5, 1, 2, 5\%$ of labeled data compared to supervised baseline.

We highlight the contributions of this paper as follows:

- By analyzing object detectors trained with limited-supervision, we identify that the nature of class-imbalance in object detection tasks impedes the effectiveness of pseudo-labeling method on SS-OD task.

- We thus proposed a simple yet effective method, Unbiased Teacher, to address the pseudo-labeling bias issue caused by class-imbalance existing in ground-truth labels and the overfitting issue caused by the scarcity of labeled data.

- Our Unbiased Teacher achieves state-of-the-art performance on SS-OD across *COCO-standard*, *COCO-additional*, and *VOC* datasets. We also provide an ablation study to verify the effectiveness of each proposed component.

## 2 RELATED WORKS

**Semi-Supervised Learning.** The majority of the recent SSL methods typically consist of (1) input augmentations and perturbations, and (2) consistency regularization. They regularize the model to be invariant and robust to certain augmentations on the input, which requires the outputs given the original and augmented inputs to be consistent. For example, existing approaches apply convention data augmentations (Berthelot et al., 2019; Laine & Aila, 2017; Sajjadi et al., 2016; Tarvainen &

---

[2]Note that there have been many works that leverages EMA, *e.g.,* ADAM optimization (Kingma & Ba, 2015), Batch Normalization (Ioffe & Szegedy, 2015), self-supervised learning (He et al., 2020; Grill et al., 2020), and SSL image classification (Tarvainen & Valpola, 2017). We, for the first time, show its effectiveness in combating class imbalance issues and detrimental effect of pseudo-labels for the object detection task.

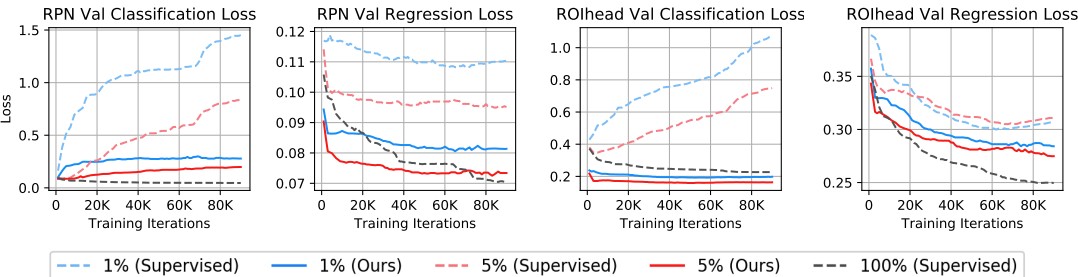

Figure 2: Validation Losses of our model and the model trained with labeled data only. When the labeled data is insufficient (1% and 5%), RPN and ROIhead classifiers suffer from overfitting, while RPN and ROIhead regression do not suffer from overfitting. Our model can significantly alleviates the overfitting issue in classifiers and also improves the validation box regression loss.

Valpola, 2017) to generate different transformations of the semantically identical images, perturb the input images along the adversarial direction (Miyato et al., 2018; Yu et al., 2019), utilize multiple networks to generate various views of the same input data (Qiao et al., 2018), mix input data to generate augmented training data and labels (Zhang et al., 2018; Yun et al., 2019; Guo et al., 2019; Hendrycks et al., 2020), or learn augmented prototypes in feature space instead of the image space (Kuo et al., 2020). However, the complexities in architecture design of object detectors hinder the transfer of existing semi-supervised techniques from image classification to object detection.

**Semi-Supervised Object Detection.** Object detection is one of the most important computer vision tasks and has gained enormous attention (Lin et al., 2017a; He et al., 2017; Redmon & Farhadi, 2017; Liu et al., 2016). While existing works have made significant progress over the years, they have primarily focused on training object detectors with fully-labeled datasets. On the other hand, there exist several semi-supervised object detection works that focus on training object detector with a combination of labeled, weakly-labeled, or unlabeled data. This line of work began even before the resurgence of deep learning (Rosenberg et al., 2005). Later, along with the success of deep learning, Hoffman et al. (2014) and Gao et al. (2019) trained object detectors on data with bounding box labels for some classes and image-level class labels for other classes, enabling detection for categories that lack bounding box annotations. Tang et al. (2016) adapted the image-level classifier of a weakly labeled category (no bounding boxes) into a detector via similarity-based knowledge transfer. Misra et al. (2015) exploited a few sparsely labeled objects and bounding boxes in some video frames and localized unknown objects in the following videos.

Unlike their settings, we follow the standard SSL setting and adapt it to the object detection task, in which the training contains a small set of labeled data and another set of completely unlabeled data (*i.e.,* only images). In this setting, Jeong et al. (2019) proposed a consistency-based method, which enforces the predictions of an input image and its flipped version to be consistent. Sohn et al. (2020b) pre-trained a detector using a small amount labeled data and generates pseudo-labels on unlabeled data to fine-tune the pre-trained detector. Their pseudo-labels are generated only once and are fixed through out the rest of training. While they can improve the performance against the model trained on labeled data, imbalance issue is not considered in existing SS-OD works. In contrast, our method not only improve the pseudo-label generation model via teacher-student mutual learning regimen (Sec. 3.2) but address the crucial imbalance issue in generated pseudo-labels (Sec. 3.3).

## 3 UNBIASED TEACHER

**Problem definition.** Our goal is to address object detection in a semi-supervised setting, where a set of labeled images $D_s = \{x_i^s, y_i^s\}_{i=1}^{N_s}$ and a set of unlabeled images $D_u = \{x_i^u\}_{i=1}^{N_u}$ are available for training. $N_s$ and $N_u$ are the number of supervised and unsupervised data. For each labeled image $x^s$, the annotations $y^s$ contain locations, sizes, and object categories of all bounding boxes.

**Overview.** As shown in Figure 3, our Unbiased Teacher consists of two training stages, the **Burn-In** stage and the **Teacher-Student Mutual Learning** stage. In the Burn-In stage (Sec. 3.1), we simply train the object detector using the available supervised data to initialize the detector. At the beginning of the Teacher-Student Mutual Learning stage (Sec. 3.2), we duplicate the initialized detector into two models (*Teacher* and *Student* models). Our Teacher-Student Mutual Learning stage aims at evolving both *Teacher* and *Student* models via a mutual learning mechanism, where

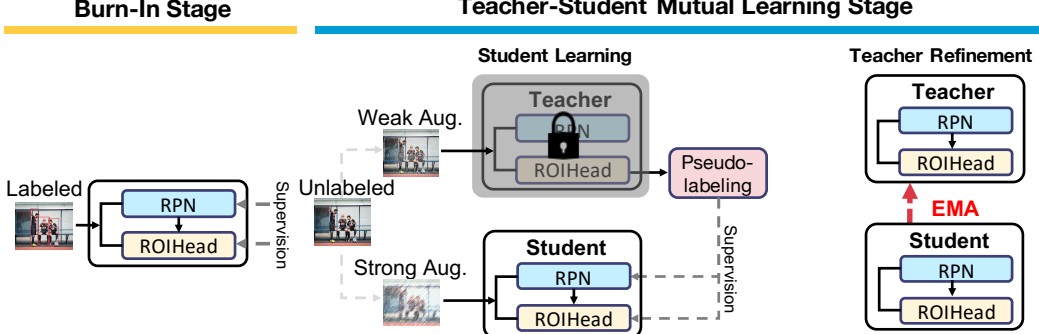

Figure 3: Overview of ***Unbiased Teacher***. Unbiased Teacher consists of two stages. ***Burn-In***: we first train the object detector using available labeled data. ***Teacher-Student Mutual Learning*** consists of two steps. **Student Learning**: the fixed teacher generates pseudo-labels to train the Student, while Teacher and Student are given weakly and strongly augmented inputs, respectively. **Teacher Refinement**: the knowledge that the Student learned is then transferred to the slowly progressing Teacher via exponential moving average (EMA) on network weights. When the detector is trained until converge in the Burn-In stage, we switch to the Teacher-Student Mutual Learning stage.

the *Teacher* generates pseudo-labels to train the *Student*, and the *Student* updates the knowledge it learned back to the *Teacher*; hence, the pseudo-labels used to train the *Student* itself are improved. Lastly, there exists class-imbalance and foreground-background imbalance problems in object detection, which impedes the effectiveness of semi-supervised techniques of image classification (*e.g.,* pseudo-labeling) being used directly on SS-OD. Therefore, in Sec. 3.3, we also discuss how Focal loss (Lin et al., 2017b) and EMA training alleviate the imbalanced pseudo-label issue.

## 3.1 BURN-IN

It is important to have a good initialization for both Student and Teacher models, as we will rely on the Teacher to generate pseudo-labels to train the Student in the later stage. To do so, we first use the available supervised data to optimize our model $\theta$ with the supervised loss $\mathcal{L}_{sup}$. With the supervised data $\boldsymbol{D}_s = \{\boldsymbol{x}_i^s, \boldsymbol{y}_i^s\}_{i=1}^{N_s}$, the supervised loss of object detection consists of four losses: the RPN classification loss $\mathcal{L}_{cls}^{rpn}$, the RPN regression loss $\mathcal{L}_{reg}^{rpn}$, the ROI classification loss $\mathcal{L}_{cls}^{roi}$, and the ROI regression loss $\mathcal{L}_{reg}^{roi}$ (Ren et al., 2015),

$$\mathcal{L}_{sup} = \sum_i \mathcal{L}_{cls}^{rpn}(\boldsymbol{x}_i^s, \boldsymbol{y}_i^s) + \mathcal{L}_{reg}^{rpn}(\boldsymbol{x}_i^s, \boldsymbol{y}_i^s) + \mathcal{L}_{cls}^{roi}(\boldsymbol{x}_i^s, \boldsymbol{y}_i^s) + \mathcal{L}_{reg}^{roi}(\boldsymbol{x}_i^s, \boldsymbol{y}_i^s). \tag{1}$$

After Burn-In, we duplicate the trained weights $\theta$ for both the Teacher and the Student models ($\theta_t \leftarrow \theta, \theta_s \leftarrow \theta$). Starting from this trained detector, we further utilize the unsupervised data to improve the object detector via the following proposed training regimen.

## 3.2 TEACHER-STUDENT MUTUAL LEARNING

**Overview.** To leverage the unsupervised data, we introduce the Teacher-Student Mutual Learning regimen, where the Student is optimized by using the pseudo-labels generated from the Teacher, and the Teacher is updated by gradually transferring the weights of continually learned Student model. With the interaction between the Teacher and the Student, both models can evolve jointly and continuously to improve detection accuracy. With the improvement on detection accuracy, this also means that the Teacher generates more accurate and stable pseudo-labels, which we identify as one of the keys for large performance improvement compared to existing work (Sohn et al., 2020b). In another perspective, we can also regard the Teacher as the temporal ensemble of the Student models in different time steps. This aligns our observation that the accuracy of the Teacher is consistently higher than the Student. As noted in prior works (Tarvainen & Valpola, 2017; Xie et al., 2020), one crucial factor in improving the Teacher model is the diversity of Student models; we thus use the

strongly augmented images as as input of the Student, but we use the weakly augmented images as input of the Teacher to provide reliable pseudo-labels.

**Student Learning with Pseudo-Labeling.** To address the lack of ground-truth labels for unsupervised data, we adapt the pseudo-labeling method to generate labels for training the Student with unsupervised data. This follows the principle of existing successful examples in semi-supervised image classification task (Lee, 2013; Sohn et al., 2020a). Similar to classification-based methods, to prevent the consecutively detrimental effect of noisy pseudo-labels (*i.e.,* confirmation bias or error accumulation), we first set a confidence threshold $\delta$ of predicted bounding boxes to filter low-confidence predicted bounding boxes, which are more likely to be false positive samples.

While the confidence threshold method have achieved tremendous success in the image classification, it is however not sufficient for object detection. This is because there also exist duplicated box predictions and imbalanced prediction issues in the SS-OD (we leave the discussion of the imbalanced prediction issue in Sec. 3.3). To address the duplicated boxes prediction issue, we remove the repetitive predictions by applying class-wise non-maximum suppression (NMS) before the use of confidence thresholding as performed in STAC (Sohn et al., 2020b).

In addition, noisy pseudo-labels can affect the pseudo-label generation model (Teacher). As a result, we detach the Student and the Teacher. To be more specific, after obtaining the pseudo-labels from the Teacher, only the learnable weights of the Student model is updated via back-propagation.

$$\theta_s \leftarrow \theta_s + \gamma \frac{\partial(\mathcal{L}_{sup} + \boldsymbol{\lambda}_u \mathcal{L}_{unsup})}{\partial \theta_s}, \quad \mathcal{L}_{unsup} = \sum_i \mathcal{L}_{cls}^{rpn}(\boldsymbol{x}_i^u, \hat{\boldsymbol{y}}_i^u) + \mathcal{L}_{cls}^{roi}(\boldsymbol{x}_i^u, \hat{\boldsymbol{y}}_i^u) \quad (2)$$

Note that we do not apply unsupervised losses for the bounding box regression since the naive confidence thresholding is not able to filter the pseudo-labels that are potentially incorrect for bounding box regression (because the confidence of predicted bounding boxes only indicate the confidence of predicted object categories instead of the quality of bounding box locations (Jiang et al., 2018)).

**Teacher Refinement via Exponential Moving Average.** To obtain more stable pseudo-labels, we apply EMA to gradually update the Teacher model. The slowly progressing Teacher model can be regarded as the ensemble of the Student models in different training iterations.

$$\theta_t \leftarrow \alpha\theta_t + (1 - \alpha)\theta_s. \quad (3)$$

This approach has been shown to be effective in many existing works, *e.g.,* ADAM optimization (Kingma & Ba, 2015), Batch Normalization (Ioffe & Szegedy, 2015), self-supervised learning (He et al., 2020; Grill et al., 2020), and SSL image classification (Tarvainen & Valpola, 2017), while we, for the first time, demonstrate its effectiveness *also* in alleviating pseudo-labeling bias issue for SS-OD (see next section).

### 3.3 BIAS IN PSEUDO-LABEL

Ideally, the methods based on pseudo-labels can address problems caused by the scarcity of labels, yet the inherent nature of imbalance in object detection tasks/datasets impedes the effectiveness of pseudo-labeling methods. As mentioned in (Oksuz et al., 2020), in object detection, there exists foreground-background imbalance (*e.g.,* background instances accounts for 70% of all training instances) and foreground classes imbalance (*e.g.,* human instances accounts for 30% of all foreground training instances in MS-COCO (Lin et al., 2014)). If standard cross-entropy is applied in the condition of insufficient training data, the model is likely prone to predict the dominant classes. This makes the prediction bias toward prevailing classes and leads to the class-imbalance issue in generated pseudo-labels. Relying on the biased pseudo-labels during training makes the imbalanced prediction issue even more severe. To address the imbalance issue in object detection, existing works have proposed several methods (Shrivastava et al., 2016; Lin et al., 2017b; Li et al., 2020).

In this work, we consider a simple yet effective method; we replace the standard cross-entropy with the multi-class Focal loss (Lin et al., 2017b) for the multi-class classification of ROIhead classifier (*i.e.,* $\mathcal{L}_{cls}^{roi}$). Focal loss is designed to put more loss weights on the samples with lower-confidence instances. As a result, it makes the model focus on hard samples, instead of the easier examples that are likely from dominant classes. Although the Focal loss is not widely used for vanilla supervised object detection settings (the accuracy of YOLOv3 (Redmon & Farhadi, 2018) even drops if the focal loss is applied), we argue that it is crucial for SS-OD due to the issue of biased pseudo-labels.

Table 1: Experimental results on *COCO-standard* comparing with CSD (Jeong et al., 2019) and STAC (Sohn et al., 2020b). *: we implement the CSD method and adapt it on the MS-COCO dataset. The results of $0.5\%$ with STAC is from their released code.

| | COCO-standard | | | | |
| | 0.5% | 1% | 2% | 5% | 10% |
|---|---|---|---|---|---|
| Supervised | $6.83 \pm 0.15$ | $9.05 \pm 0.16$ | $12.70 \pm 0.15$ | $18.47 \pm 0.22$ | $23.86 \pm 0.81$ |
| CSD* | $7.41 \pm 0.21$ (+0.58) | $10.51 \pm 0.06$ (+1.46) | $13.93 \pm 0.12$ (+1.23) | $18.63 \pm 0.07$ (+0.16) | $22.46 \pm 0.08$ (-1.40) |
| STAC | $9.78 \pm 0.53$ (+2.95) | $13.97 \pm 0.35$ (+4.92) | $18.25 \pm 0.25$ (+5.55) | $24.38 \pm 0.12$ (+5.86) | $28.64 \pm 0.21$ (+4.78) |
| Unbiased Teacher | $\mathbf{16.94 \pm 0.23}$ (+10.11) | $\mathbf{20.75 \pm 0.12}$ (+11.72) | $\mathbf{24.30 \pm 0.07}$ (+11.60) | $\mathbf{28.27 \pm 0.11}$ (+9.80) | $\mathbf{31.50 \pm 0.10}$ (+7.64) |

On the other hand, we also observe that the EMA training can also alleviate the imbalanced pseudo-labeling biased issue due to the conservative property of the EMA training. To be more specific, with the EMA mechanism, the new Teacher model is regularized by the previous Teacher model, and this prevents the decision boundary from drastically moving toward the minority classes. In detail, the weights of the Teacher model can be represented as follows:

$$\theta_t^i = \hat{\theta} - \gamma \sum_{k=1}^{i-1} (1 - \alpha^{-k+(i-1)}) \frac{\partial(\mathcal{L}_{sup} + \boldsymbol{\lambda}_u \mathcal{L}_{unsup})}{\partial \theta_s^k}, \tag{4}$$

where $\hat{\theta}$ is the model weight after the burn-in stage, $\theta_t^i$ is the Teacher model weight in $i$-th iteration, $\theta_s^k$ is the Student model weight in $k$-th iteration, $\gamma$ is the learning rate, and $\alpha$ is the EMA coefficient.

The regularization of the previous Teacher model is equivalent to putting an additional small coefficient on the gradients on Student models in previous steps. With the slowly altered decision boundary (*i.e.,* higher stability), the pseudo-labels of these unlabeled instances are less likely to change dramatically, and this prevents the decision boundary from moving toward minority classes (*i.e.,* majority class bias). Thus, the EMA-trained Teacher model is beneficial for producing more stable pseudo-labels and addressing the class-imbalance issue in SS-OD.

We note that the class-imbalance issue is crucial when using pseudo-labeling method to address semi-supervised or other low-label object detection tasks. There indeed exist other class-imbalance methods that can potentially improve the performance, but we leave this for future research.

## 4 EXPERIMENTS

**Datasets.** We benchmark our proposed method on experimental settings using MS-COCO (Lin et al., 2014) and PASCAL VOC (Everingham et al., 2010) following existing works (Jeong et al., 2019; Sohn et al., 2020b). Specifically, there are three experimental settings: (1) *COCO-standard*: we randomly sample 0.5, 1, 2, 5, and 10% of labeled training data as a labeled set and use the rest of the data as the training unlabeled set. (2) *COCO-additional*: we use the standard labeled training set as the labeled set and the additional *COCO2017-unlabeled* data as the unlabeled set. (3) *VOC*: we use the VOC07 *trainval* set as the labeled training set and the VOC12 *trainval* set as the unlabeled training set. Model performance is evaluated on the VOC07 test set.

**Implementation Details.** For a fair comparison, we follow STAC (Sohn et al., 2020b) to use Faster-RCNN with FPN (Lin et al., 2017a) and ResNet-50 backbone (He et al., 2016) as our object detectior, where the feature weights are initialized by the ImageNet-pretrained model, same as existing works (Jeong et al., 2019; Sohn et al., 2020b). We use confidence threshold $\delta = 0.7$. For the data augmentation, we apply random horizontal flip for weak augmentation and randomly add color jittering, grayscale, Gaussian blur, and cutout patches for strong augmentations. Note that we do not apply any geometric augmentations, which are used in STAC. We use $AP_{50:95}$ (denoted as mAP) as evaluation metric, and the performance is evaluated on the Teacher model. More training and implementation details can be found in the Appendix.

### 4.1 RESULTS

**COCO-standard.** We first evaluate the efficacy of our Unbiased Teacher on COCO-standard (Table 1). When there are only $0.5\%$ to $10\%$ of data labeled, our model consistently performs favorably

Table 2: Experimental results on *COCO-additional* comparing with CSD (Jeong et al., 2019) and STAC (Sohn et al., 2020b). *: we implement the CSD method and adapt it on the MS-COCO dataset. Note that 1x represents 90K training iterations, and $N$x represents $N{\times}90$K training iterations.

| | COCO-additional | | | | |
|---|---|---|---|---|---|
| | Supervised (1x) | Supervised (3x) | CSD (3x) | STAC (6x) | Ours (3x) |
| $AP^{50:95}$ | 37.63 | 40.20 | 38.82 | 39.21 | 41.30 |

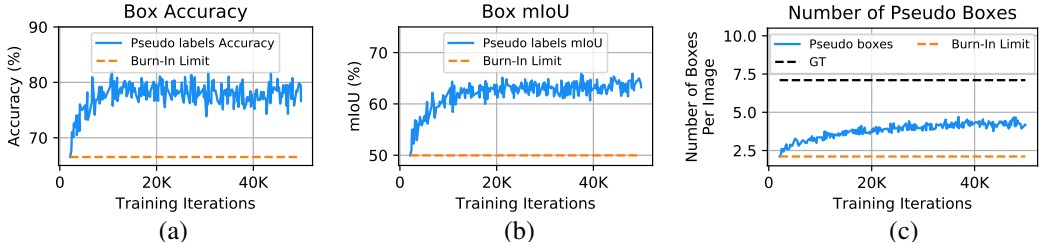

(a)  (b)  (c)

Figure 4: Pseudo-label improvement on (a) accuracy, (b) mIoU, and (c) number of bounding boxes in the case of *COCO-standard* 1% labeled data. We measure the (a) accuracy and (b) mIoU by comparing the ground-truth boxes and pseudo boxes. The Burn-In limit curves indicate the pseudo-boxes obtained from the model right after the Burn-In stage without further refinement (*i.e.,* the model trained on labeled data only). GT curve on the number of boxes figure indicates the averaged number of bounding boxes in the GT labels, and we showed that there are around 7 bounding boxes per image on average in MS-COCO. This result indicates our model can generate more accurate pseudo-labels after the Burn-In stage (*i.e.,* 2k iterations).

against the state-of-the-art methods, CSD (Jeong et al., 2019) and STAC (Sohn et al., 2020b). It is worth noting that our model trained on 1% labeled data achieves 20.75% mAP, which is even higher than STAC trained on 2% labeled data (mAP 18.25%), CSD trained on 5% labeled data (mAP 18.57%), and the supervised baseline trained on 5% labeled data (mAP 18.47%). We also observe that, as there are less labeled data, the improvements between our method and the existing approaches becomes larger. Unbiased Teacher consistently shows around 10 absolute mAP improvements when using less than 5% of labeled data compared to supervised method. We attribute the improvements to several crucial factors:

1) *More accurate pseudo-labels*. When leveraging the pseudo-labeling and consistency regularization between two networks (Teacher and Student in our case), it is critical to make sure pseudo-labels are accurate and reliable. Existing method attempts to do this by training the pseudo-label generation model using all the available labeled data and is completely frozen afterwards. In contrast, in our framework, our pseudo-label generation model (Teacher) continues to evolve gradually and smoothly via Teacher-Student Mutual Learning. This enables the Teacher to generate more accurate pseudo-labels as presented in Figure 4, which are properly exploited in the training of the Student.

2) *Class-imbalance on pseudo-labels*. Our improvement also comes from both the use of the EMA and the Focal loss (Lin et al., 2017b), which addresses the class-imbalanced pseudo-labeling issue. As mentioned in Sec. 3.3, using more balanced pseudo-labels not only avoids the consecutive biased prediction problem but also benefits the predictions on the minority classes. Later in Sec. 4.2, we present the details of the ablation study on the EMA and the Focal loss.

**COCO-additional and VOC.** In the previous section, we presented Unbiased Teacher can successfully leverage very small amounts of labeled data. We now aim to verify whether the model trained on 100% supervised data can be further improved by using additional unlabeled data. We thus consider *COCO-additional* and *VOC* and present the results in Table 1 and 3.

In the case of *COCO-additional* (Table 2), compared with supervised only model, our model has a 1.10 absolute AP improvement. We also found a similar trend in the *VOC* experiment (Table 3). With *VOC07* as labeled set and *VOC12* as an additional unlabeled set, STAC shows 2.51 absolute mAP improvement with respect to the supervised model, whereas our model demonstrates 6.56 ab-

Table 3: Results on *VOC* comparing with CSD (Jeong et al., 2019) and STAC (Sohn et al., 2020b).

| | Backbone | Labeled | Unlabeled | $AP_{50}$ | $AP_{50:95}$ |
|---|---|---|---|---|---|
| Supervised (from Ours) | ResNet50-FPN | VOC07 | None | 72.63 | 42.13 |
| CSD | ResNet101-R-FCN | | VOC12 | 74.70 (+2.07) | - |
| STAC | ResNet50-FPN | VOC07 | VOC12 | 77.45 (+4.82) | 44.64 (+2.51) |
| Unbiased Teacher | ResNet50-FPN | | | 77.37 (+4.74) | **48.69** (+6.56) |
| CSD | ResNet101-R-FCN | | VOC12 | 75.10 (+2.47) | - |
| STAC | ResNet50-FPN | VOC07 | + | 79.08 (+6.45) | 46.01 (+3.88) |
| Unbiased Teacher | ResNet50-FPN | | *COCO20cls* | 78.82 (+6.19) | **50.34** (+8.21) |

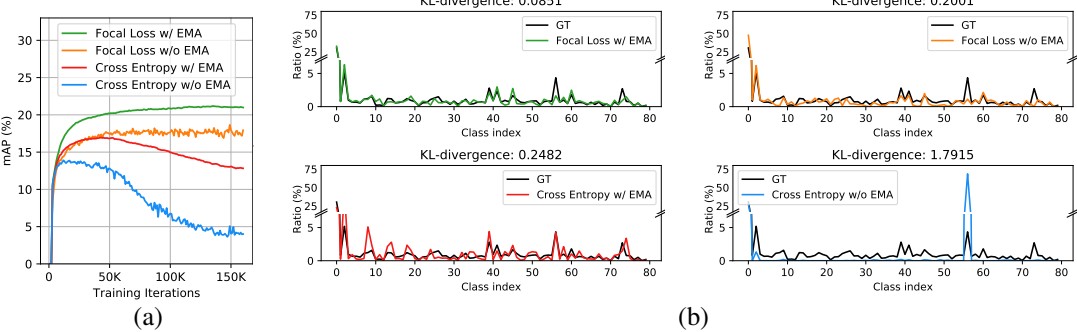

(a)                         (b)

Figure 5: Ablation study on the EMA and the Focal loss in the case of *COCO-standard* 1% labeled data. (a) mAP of the models using the Focal loss or cross-entropy and applying the EMA or standard training. (b) Class empirical distribution (*i.e.,* histogram) of pseudo-labels generated by each model and compute $\mathcal{KL}$-divergence between the ground-truth labels distribution and the pseudo-label distribution. Among these models, the model using the Focal loss and EMA training (*i.e.,* green curve) achieves the best mAP with the most balanced pseudo-labels .

solute mAP improvement. To further examine whether increasing the size of unlabeled data can further improve the performance, we follow CSD and STAC to use *COCO20cls* dataset[3] as an additional unlabeled set. STAC shows 3.88 absolute mAP improvement, while our model achieves 8.21 absolute mAP improvement. These results demonstrate that our model can further improve the object detector trained on existing labeled datasets by using more unlabeled data. Note that, following STAC, we use a more challenging metric, $AP_{50:95}$, which averages the ten values over $AP_{50}$ to $AP_{95}$ since the metric of $AP_{50}$ has been indicated as a saturated metric by the prior work (Cai & Vasconcelos, 2018; Sohn et al., 2020b).

## 4.2 ABLATION STUDY

**Effect of the EMA training.** We first examine the effect of EMA training and present a comparison between our model with EMA and without EMA. Our model without EMA is where the model weights of Teacher and Student are shared during the training stage, and it implies the Teacher model is also updated when the student model is optimized by using unlabeled data and pseudo-labels. Note that the state-of-the-art semi-supervised classification model, FixMatch (Sohn et al., 2020a) similarly shares the model weights of the Teacher and the Student models.

From Figure 5, we observe that our model with EMA is superior to without EMA, and this trend can be found both in the model using the Focal loss and cross-entropy. To further analyze the diverged results, we visualize the class distribution of pseudo-labels generated by each model and measure the $\mathcal{KL}$-divergence between the ground-truth labels distribution and the pseudo-labels distribution. With the use of cross-entropy and standard training (*i.e.,* without EMA training), the model generates the imbalanced pseudo-labels. To be more specific, the instances of most object categories in pseudo-labels disappear, while only instances of specific object categories remain. We observe that using the EMA training can alleviate the imbalanced pseudo-labels issue and reduces the $\mathcal{KL}$-divergence

---

[3]*COCO20cls* is generated by only leaving COCO images which have object categories that overlap with the object categories used in PASCAL VOC07.

from 1.7915 to 0.2482. On the other hand, we also observe that the model with EMA has a smoother learning curve compared with the model without EMA. This is because the model weight of the pseudo-label generation model (Teacher) is detached from the optimized model (Student). The pseudo-label generation model can thus prevent the detrimental effect caused by the noisy pseudo-labels (*e.g.,* false positive boxes) as we describe in Sec. 3.2.

In sum, the EMA training has several advantages: it 1) prevents the imbalanced pseudo-labels issue caused by the imbalanced nature in low-labeled object detection tasks, 2) prevents the detrimental effect caused by the noisy pseudo-labels, and 3) the Teacher model can be regarded as the temporal ensembles model of Student models in different time steps.

**Effect of the Focal loss.** In addition to the EMA training, we also verify the effectiveness of the Focal loss. As presented in Figure 5, the model using Focal loss can perform favorably against the model using cross-entropy. The model trained with the Focal loss can generate the pseudo-label which distribution is more similar to the distribution of ground-truth labels, and it can improve the $\mathcal{KL}$-divergence from 1.7915 (Cross entropy w/o EMA) to 0.2001 (Focal loss w/o EMA) and mAP from 13.42 to 17.85. When EMA training is applied, the $\mathcal{KL}$-divergence of the model with the Focal loss can be further improved from 0.2482 (Cross entropy w/ EMA) to 0.0851 (Focal loss w/ EMA) and mAP improve from 16.91 to 21.19. This confirms the effectiveness of the Focal loss in handling the class imbalance issues existed in the semi-supervised object detection. The reduction of $\mathcal{KL}$-divergence (*i.e.,* better-fitting pseudo-label distributions to ground-truth label distributions) results in the mAP improvement.

**Other ablation studies.** We also ablate the effects of the Burn-In stage, pseudo-labeling thresholding, EMA rates, and unsupervised loss weights in the Appendix.

## 5 CONCLUSION

In this paper, we revisit the semi-supervised object detection task. By analyzing the object detectors in low-labeled scenarios, we identify and address two major issues: overfitting and class imbalance. We proposed Unbiased Teacher — a unified framework consisting of a Teacher and a Student that jointly learn to improve each other. In the experiments, we show our model prevents pseudo-labeling bias issue caused by class imbalance and overfitting issue due to labeled data scarcity. Our Unbiased Teacher achieves satisfactory performance across multiple semi-supervised object detection datasets.

## 6 ACKNOWLEDGMENTS

Yen-Cheng Liu and Zsolt Kira were partly supported by DARPA's Learning with Less Labels (LwLL) program under agreement HR0011-18-S-0044, as part of their affiliation with Georgia Tech.

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

# A APPENDIX

## A.1 EMA ON IMBALANCED PSEUDO-LABELING ISSUE

To empirically examine the effectiveness of EMA on imbalance, we present the pseudo-label distribution in different training iterations as presented in Figure 6. At the beginning of training (*i.e.*, 30k), both Teacher models with and without EMA could generate the balanced pseudo-labels (the KL divergence between ground-truth labels and pseudo-labels are both small). However, since the Student model is trained with the pseudo-labels generated by the Teacher models, the model without EMA starts biasing towards specific classes. In contrast, with the EMA training, the model generates less imbalanced pseudo-labels. Note that, although the EMA is applied, the balance issue still exists. We thus apply Focal loss to enhance the ability to mitigate the imbalance issue further.

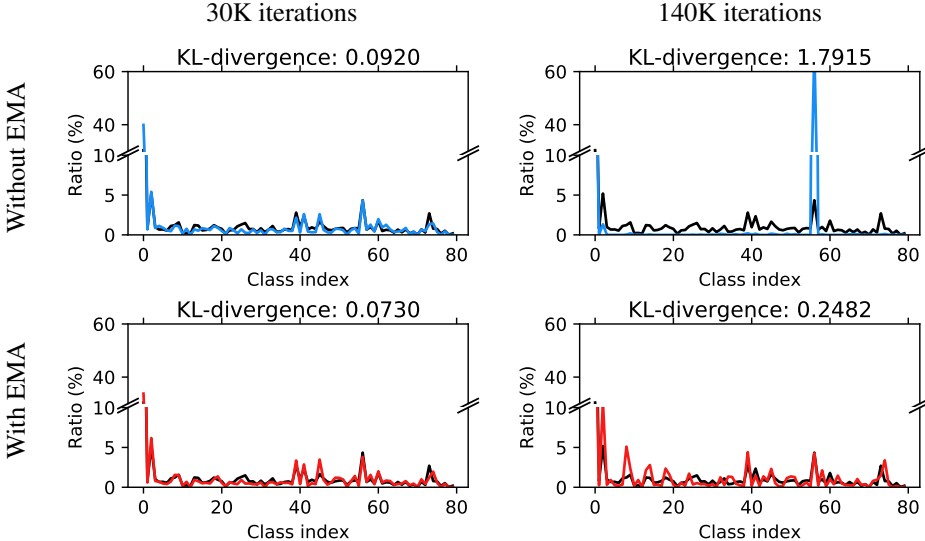

Figure 6: Ablation study on EMA at different training iterations. Both the models with EMA and without EMA have pseudo-label distributions, which are similar to the ground-truth distributions in the early stage of training iterations. However, the model without EMA tends to generate more biased pseudo-label distribution later during training.

## A.2 ADDITIONAL ABLATION STUDY

In addition to the ablation studies provided in the main paper, we further ablate Unbiased Teacher in the following sections.

### A.2.1 EFFECT OF BURN-IN STAGE

As mentioned in Section 3.1, it is crucial to have a good initialization for both *Student* and *Teacher* models. We thus present a comparison between the model with and without the Burn-In stage in Figure 7. We observe that, with the Burn-In stage, the model can derive more accurate pseudo-boxes in the early stage of the training. As a result, the model can achieve higher accuracy in the early stage of the training, and it also achieves better results when the model is converged.

### A.2.2 EFFECT OF PSEUDO-LABELING THRESHOLD

As mentioned in Section 3.3, we apply confidence thresholding to filter these low-confidence predicted bounding boxes, which are more likely to be false-positive instances. To show the effectiveness of thresholding, we first provide the accuracy of predicted bounding boxes before and after the pseudo-labeling in Figure 8.

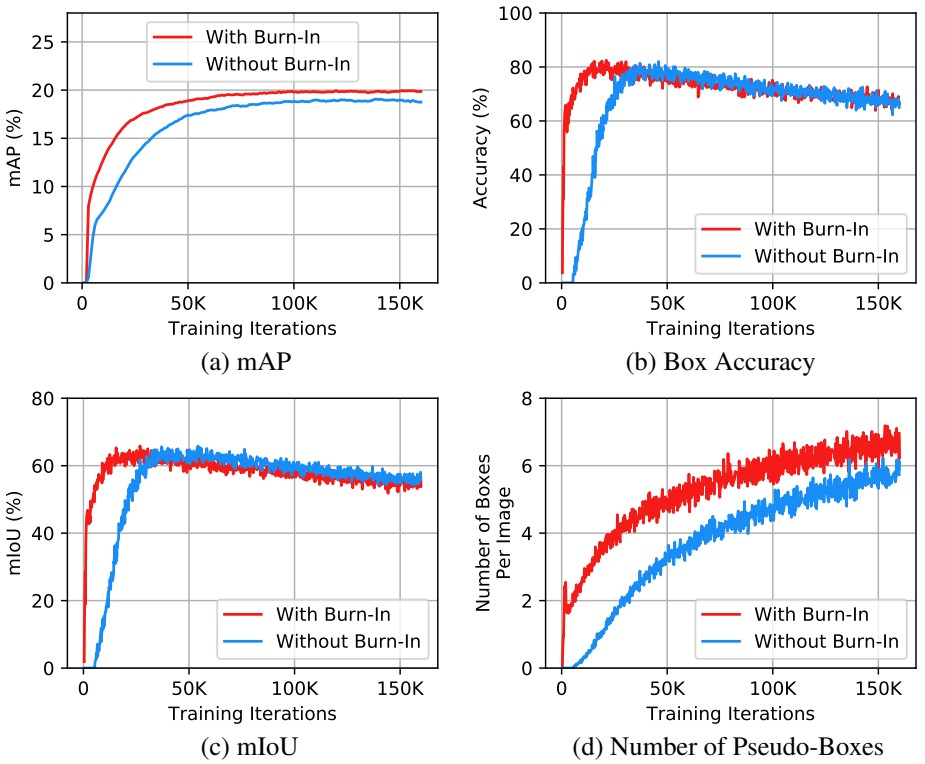

Figure 7: In the case of *COCO-standard* 1% labeled data, (a) Unbiased Teacher with Burn-In stage achieve higher mAP against Unbiased Teacher without Burn-In stage. Using Burn-In Stage results in the early improvement of (b) box accuracy and (c) mIoU. (d) Unbiased Teacher with Burn-In stage can derive more pseudo-boxes than Unbiased Teacher without Burn-In stage.

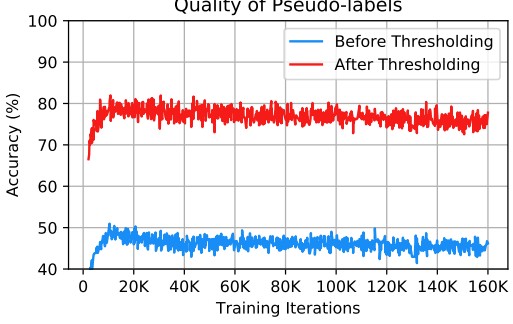

Figure 8: Pseudo-label accuracy improvement with the use of confidence thresholding. We measure the accuracy by comparing the ground-truth labels and predicted labels before and after confidence thresholding. This result indicates that confidence thresholding can significantly improve the quality of pseudo-labels.

When varying the threshold value $\delta$ from 0 to 0.9, as expected, the number of generated pseudo-boxes increases as the threshold $\delta$ reduces (Figure 9). The model using excessively high threshold (*e.g.,* $\delta = 0.9$) cannot perform satisfactory results, as the number of generated pseudo-labels is very low. On the other hand, the model using a low threshold (*e.g.,* $\delta = 0.6$) also cannot achieve favorable results since the model generates too many bounding boxes, which are likely to be false-positive instances. We also observe that the model cannot even converge if the threshold is below 0.5.

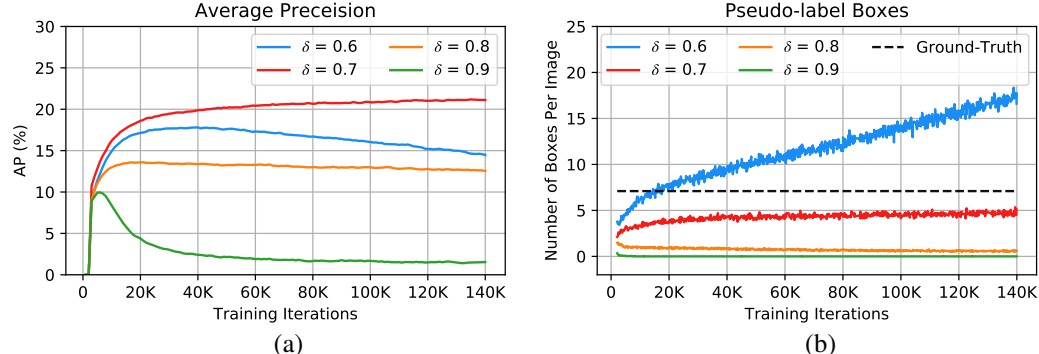

Figure 9: (a) Validation AP and (b) number of pseudo-label bounding boxes per image with various pseudo-labeling thresholds $\delta$. With an excessively low threshold (*e.g.,* $\delta = 0.6$), the model has a lower AP, as it predicts more pseudo-labeled bounding boxes compared to the number of bounding boxes in ground-truth labels. On the other hand, the performance of the model using an excessively high threshold (*e.g.,* $\delta = 0.9$) drops as it cannot predict sufficient number of bounding boxes in its generated pseudo-labels.

### A.2.3 Effect of EMA Rates

We also evaluate the model using various EMA rate $\alpha$ from $0.5$ to $0.9999$ and present the mAP result of the Teacher model in Figure 10. We observe that, with a smaller EMA rate (*e.g.,* $\alpha = 0.5$), the model has lower mAP and higher variance, as the Student contributes more to the Teacher model for each iteration. This implies the Teacher model is likely to suffer from the detrimental effect caused by noisy pseudo-labels. This unstable learning curve can be stabilized and improved as the EMA rate $\alpha$ increases. When the EMA rate $\alpha$ achieves $0.99$, it performs the best mAP. However, if the EMA rate $\alpha$ keeps increasing, the teacher model will grow overly slow as the Teacher model derive the next model weight mostly from the previous Teacher model weight.

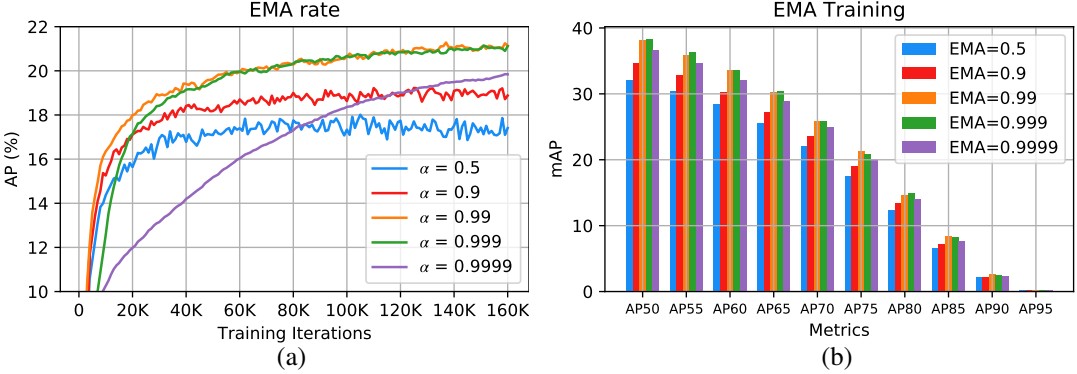

Figure 10: Validation AP on the Teacher model with various MMA rates $\alpha$. (a) With a small MMA rate (*e.g.,* $\alpha = 0.5$), the Teacher model has lower AP and larger variance. In contrast, as the MMA rate grows to $0.99$, the Teacher model can gradually improve along the training iterations. However, when the MMA grows to $0.9999$, the Teacher model grows overly slow but has lowest variance. (b) We breakdown the AP metric into APs from $AP_{50}$ to $AP_{95}$.

### A.2.4 Effect of Unsupervised Loss Weights

To examine the effect unsupervised loss weights, we vary the unsupervised loss weight $\lambda_u$ from $1.0$ to $8.0$ in the case of *COCO-standard* $10\%$ labeled data. As shown in Table 4, with a lower unsupervised loss weight $\lambda_u = 1.0$, the model performs $29.30\%$. On the other hand, we observe that the model performs the best with unsupervised loss weight $\lambda = 5.0$. However, when the weight increases to $8.0$, the training of the model cannot converge.

Table 4: Ablation study of varying unsupervised loss weight $\lambda_u$ on the model trained using $10\%$ labeled and $90\%$ unlabeled data.

| $\lambda_u$ | 1.0 | 2.0 | 4.0 | 5.0 | 6.0 | 8.0 |
|---|---|---|---|---|---|---|
| AP (%) | 29.30 | 30.64 | 31.82 | 32.00 | 31.80 | Cannot Converge |

### A.3 AP BREAKDOWN FOR COCO-STANDARD

We present an AP breakdown for COCO-standard $0.5\%$ labeled data. As mentioned in Section 4, our proposed model can perform favorably against both STAC (Sohn et al., 2020b) and CSD (Jeong et al., 2019). This trend appears in all evaluation metrics from $AP_{50}$ to $AP_{95}$, as shown in Figure 11, and it confirms that our model is preferable for handling extremely low-label scenario compared to the state of the arts.

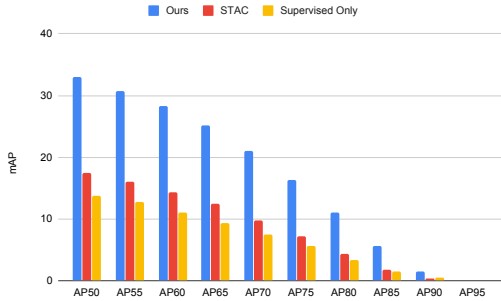

Figure 11: Evaluation metric breakdown of all methods on $0.5\%$ labeled data.

### A.4 IMPLEMENTATION AND TRAINING DETAILS

**Network and framework.** Our implementation builds upon the Detectron2 framework (Wu et al., 2019). For a fair comparison, we follow the prior work (Sohn et al., 2020b) to use Faster-RCNN with FPN (Lin et al., 2017a) and ResNet-50 backbone (He et al., 2016) as our object detection network.

**Training.** At the beginning of the Burn-In stage, the feature backbone network weights are initialized by the ImageNet-pretrained model, which is same as existing works (Jeong et al., 2019; Tang et al., 2020; Sohn et al., 2020b). We use the SGD optimizer with a momentum rate 0.9 and a learning rate 0.01, and we use constant learning rate scheduler. The batch size of supervised and unsupervised data are both 32 images. For the *COCO-standard*, we train 180k iterations, which includes $1/2/6/12/20$k iterations for $0.5\%/1\%/2\%/5\%/10\%$ in the Burn-In stage and the remaining iterations in the Teacher-Student Mutual Learning stage. For the *COCO-additional*, we train 360k iterations, which includes 90k iterations in the Burn-Up stage and the remaining 270k iterations in the Teacher-Student Mutual Learning stage.

**Hyper-parameters.** We use confidence threshold $\delta = 0.7$ to generate pseudo-labels for all our experiments, the unsupervised loss weight $\lambda_u = 4$ is applied for *COCO-standard* and *VOC*, and the unsupervised loss weight $\lambda_u = 2$ is applied for *COCO-additional*. We apply $\alpha = 0.9996$ as the EMA rate for all our experiments. Hyper-parameters used are summarized in Table 5.

**Data augmentation.** As shown in Table 6, we apply randomly horizontal flip for weak augmentation and randomly add color jittering, grayscale, Gaussian blur, and cutout patches (DeVries & Taylor, 2017) for the strong augmentation. Note that we do not apply any image-level or box-level geometric augmentations, which are used in STAC (Sohn et al., 2020b). In addition, we do not aggressively search the best hyper-parameters for data augmentations, and it is possible to obtain better hyper-parameters.

Table 5: Meanings and values of the hyper-parameters used in experiments.

| Hyper-parameter | Description | *COCO-standard* and VOC | *COCO-additional* |
|---|---|---|---|
| $\delta$ | Confidence threshold | 0.7 | 0.7 |
| $\lambda_u$ | Unsupervised loss weight | 4 | 2 |
| $\alpha$ | EMA rate | 0.9996 | 0.9996 |
| $b_l$ | Batch size for labeled data | 32 | 16 |
| $b_u$ | Batch size for unlabeled data | 32 | 16 |
| $\gamma$ | Learning rate | 0.01 | 0.01 |

Table 6: Detail of data augmentations. Probability in the table indicates the probability of applying the corresponding image process.

| Weak Augmentation | | | |
|---|---|---|---|
| Process | Probability | Parameters | Descriptions |
| Horizontal Flip | 0.5 | - | None |
| **Strong Augmentation** | | | |
| Process | Probability | Parameters | Descriptions |
| Color Jittering | 0.8 | (brightness, contrast, saturation, hue) = (0.4, 0.4, 0.4, 0.1) | Brightness factor is chosen uniformly from [0.6, 1.4], contrast factor is chosen uniformly from [0.6, 1.4], saturation factor is chosen uniformly from [0.6, 1.4], and hue value is chosen uniformly from [-0.1, 0.1]. |
| Grayscale | 0.2 | None | None |
| GaussianBlur | 0.5 | (sigma_x, sigma_y) = (0.1, 2.0) | Gaussian filter with $\sigma_x = 0.1$ and $\sigma_y = 2.0$ is applied. |
| CutoutPattern1 | 0.7 | scale=(0.05, 0.2), ratio=(0.3, 3.3) | Randomly selects a rectangle region in an image and erases its pixels. We refer the detail in Zhong et al. (2017). |
| CutoutPattern2 | 0.5 | scale=(0.02, 0.2), ratio=(0.1, 6) | Randomly selects a rectangle region in an image and erases its pixels. We refer the detail in Zhong et al. (2017). |
| CutoutPattern3 | 0.3 | scale=(0.02, 0.2), ratio=(0.05, 8) | Randomly selects a rectangle region in an image and erases its pixels. We refer the detail in Zhong et al. (2017). |

**Evaluation Metrics.** $AP_{50:95}$ is used to evaluate all methods following the prior works (Law & Deng, 2018; Sohn et al., 2020b).

