# OpenReview forum: "Unbiased Teacher for Semi-Supervised Object Detection"
_ICLR.cc/2021/Conference — ICLR 2021 Poster_

### Official Review · AnonReviewer1 · 2020-10-27
**Good paper on semi-supervised object detection**

**Rating:** 7
**Confidence:** 5

**Review:**

+This paper presents a good work on semi-supervised object detection (SSOD), which is a very challenging task. Although there are great progresses on semi-supervised classification, the SSOD is lying behind. This paper shows very good results over the supervised baselines, even when all annotations are used in COCO.

+The proposed method is very simple. It seems the paper is easy to be reproduced.

+It is a good idea to use EMA of mean teacher for SSOD. In addition, the focal loss (FL) is shown to be very useful in SSOD.

A few questions:

-The default FL is mainly for +/- class imbalance. Do you have modified for imbalance among all positive classes?

-It seems FL is more useful than EMA. However, FL is not well ablated. For example, it is the balance between +/- classes more important or among all positive classes? How about the other SSOD, e.g. STAC, using FL?

-It is understandable that EMA model is more reliable. But I don't see it can be beneficial for class imbalance issue explicitly.

-The comparisons with the baseline supervised models seem to be not very fair. For example, in the last column of Table 1, the supervised model uses 1x schedule (90k iterations), but the proposed method uses 4x schedule. From the Detectron2 github, the 3x model should have AP of 40.2. When compared with 4x supervised baseline, the gain of this paper should be much smaller. The same issue could also be in the other COCO-standard experiments. More fair comparisons should be presented.

-Don't quite understand Fig. 4. Too few description. What are burn-in limit and GT. Why are they constant?

-For the paragraph of "Class-imbalance on pseudo-labels", if you don't show results, why you write it in Sec. 4.1, instead of Sec. 4.2?

-It is very weird to cite Law & Deng, 2018, when you refer to AP50 saturation. There are tons of paper out there showing that.

-For the AP curves, do you evaluate on the full val2017 set? It seems the data points are very dense. Isn't it expensive?

-I don't think the smoother curve of EMA teacher is the weights of teacher model is detached from the student model.

=====updates========

Most of my concerns have been addressed by the rebuttal and all the other reviews are positive. l will remain my original recommendation.

---

> ### Author Response · Authors · 2020-11-18
> **We added some experiments to address the concerns about the Focal loss, and we also clarified the learning rate schedule and Figure 4**
>
> We appreciate the reviewer’s effort and thank Reviewer1 for providing constructive feedback and insightful questions. Our paper indeed has been further improved by these suggestions.
>
> **1. Do we modify the Focal loss?**
> - Yes, we modify the binary focal loss to multi-class focal loss for all positive classes and the background class. Our ROIhead predicts all foreground classes and the background class ($N+1$ nodes class prediction). We have clarified this in the Section 3.3 of the revised paper.
> ---
> **2. Ablation on variants of Focal loss**
>
> To compare the effect of foreground-background imbalance and foreground classes imbalance, we construct two variants of focal loss, foreground-only focal loss and binary focal loss. For foreground-only focal loss, we only apply focal loss weight ($1-p$) on these foreground instances and apply constant $1$ on all background instances (*i.e.,* thus becomes standard cross-entropy for background instances). We experiment these Focal loss variants on COCO-standard 1% labeled setting. We observe that foreground-only Focal loss can improve $2.7$ AP compared to the model using the cross-entropy. When the binary Focal loss is used, it could further improve $0.83$ AP. By addressing both foreground-background imbalance and foreground classes imbalance, our model could perform the best result ($20.75$ AP).
>
> |              	| Cross-entropy 	 | Foreground-only Focal loss 	 | Binary Focal loss 	  | Multi-class Focal loss 	|
> |--------------	|:-------------:	|:--------------------------:	|:-----------------:	|:----------------------:	|
> | $AP^{50:95}$ 	|    $16.95$    	|           $19.65$          	|      $20.48$      	|         $20.75$        	|
>
> ---
> **3. How does the STAC perform if the Focal loss is used?**
> - We also apply multi-class focal loss on STAC and experiment on the COCO-standard 1% labeled setting as shown in the following table. There is no improvement when we apply Focal loss on STAC. This is because the pseudo-labels generated by STAC are fixed and never refined during the training iterations, and thus STAC does not suffer from pseudo-labeling bias problems propagating across training, though the overall performance is much worse due to this limited training regime.
>
> |               	| STAC w/ Cross-entropy 	     | STAC w/ multi-class Focal loss 	    |
> |---------------	|:---------------------:    	    |:------------------------------:	    |
> | $AP^{50:95}$ 	|         $13.97$             	|             $ 13.94$                 	|
>
> ---

---

> > ### Author Response · Authors · 2020-11-18
> > **Cont'd**
> >
> > **5.Learning rate scheduler**
> > - Thank you for bringing this up. For the COCO-additional experiments, we used the reported value from STAC paper. For a fair and better comparison, we show the complete results for COCO-additional in the following table. Our model achieves $41.30$ when $3$x schedule is used, and it performs favorably against CSD and STAC. Note that supervised-only baseline benefits from weight-decay learning scheduler, while our Unbiased Teacher only used a simple constant learning rate (*i.e.,* $0.01$) through whole training. We believe the Unbiased Teacher could be further improved by using a more advanced learning rate scheduler, but we leave this for future research. We appreciate the Reviewer 1 for pointing this out, and we have corrected this table accordingly in the revised paper (Table 2 of the revised paper).
> > |                            	| Schedule (x = $90K$)          | $AP^{50:95}$ 	|
> > |----------------------------	|:----------------------------------------:	|:-------:	|
> > |      Supervised Only       	|        $1$x        	|  $37.63$  	|
> > |      Supervised Only       	|         $3$x         	|  $40.20$  	|
> > |    CSD (our reproduced)    	|         $3$x         	|  $38.82$  	|
> > | STAC (from original paper) 	|         $6$x         	|  $39.21$  	|
> > |            Ours            	|         $3$x         	|  $41.30$  	|
> > |            Ours            	|         $4$x         	|  $41.53$  	|
> >
> > - For the COCO-standard setting, all methods in Table 1 use a $2$x scheduler (*i.e.,* $180K$ iterations for training) for the fair comparison. We also observed that the supervised-only baseline converges in the early stage of the training due to the severe overfitting issue of classifiers as presented in Figure 1 of the main paper.
> > ---
> > **6. Clarification on Figure 4**
> > - Thank you for the question; we have revised the paper to better explain this. We intended to show that the teacher-student mutual learning stage could further improve the quality of pseudo-labels (in terms of pseudo-box accuracy, pseudo-box IoU, and number of pseudo-boxes) compared to the model.
> > - The burn-in limit curves indicate the pseudo-boxes obtained from the model right after the burn-in stage without further refinement (*i.e.,* the model trained on labeled data only). GT curve on the number of boxes figure indicates the averaged number of bounding boxes in the ground-truth labels, and we showed that there are around $7$ bounding boxes per MS-COCO image on average. We have added further detail to the revision to make this clear.
> >
> > ---
> > **7. Paragraph of “Class-imbalance on pseudo-labels” in Section 4.1**
> > - Thanks for the suggestion. We put the description in Section 4.1 to summarize the reasons why we could improve in comparison to existing works. We have re-organized this paragraph accordingly.
> > ---
> > **8. Reference of the saturated evaluation metric**
> > - Thanks for the suggestions. We add [1, 2] to refer to $AP^{50}$ saturation. We are happy to cite other papers if Reviewer 1 has other alternatives.
> >
> > [1] “Cascade R-CNN: Delving into High Quality Object Detection”, Cai et al., CVPR 2018\
> > [2] “A Simple Semi-Supervised Learning Framework for Object Detection”, Sohn et al., arXiv 2020
> >
> > ---
> > **9.Do we evaluate on the full val2017 set?**
> > - Yes, we evaluate on the full val2017 set. We evaluate all models on full val2017 set for every 1k iterations.

---

> > ### Author Response · Authors · 2020-11-18
> > **Cont'd**
> >
> > **4. How does EMA alleviate biased pseudo-labeling issues?**
> >
> > - The reasons why EMA can alleviate biased pseudo-labeling issues are two-fold:
> >     - In the teacher refinement stage, EMA **improves the stability (*i.e.,* consistent balance) of the teacher model** by preserving the previous teacher model weights.
> >     - In the student learning stage, EMA **makes the prediction of the student model more balanced** by using more stable targets (*e.g.,* pseudo-labels) from a more balanced teacher.
> > - **What if EMA is not used?** When EMA is not used and the teacher model does not preserve the previous teacher model weights, the teacher’s prediction will gradually bias to the majority class. This is because biasing prediction toward the majority classes enables the model to minimize the training loss by easily predicting the majority classes for most instances. Thus, the new decision boundary of the teacher model moves towards the minority classes (*i.e.* pushes into the feature space of the minority class, hence causing the model to classify them as the majority classes). As the pseudo-labeling method uses the previous teacher’s prediction as another supervision for the student in the next step, the pseudo-labeling, which is designed to address low-label scenarios, aggravates the imbalanced prediction issue.
> >
> > - **Higher stability of teacher model:** With the EMA mechanism, the new teacher model is regularized by the previous teacher model, and this prevents the decision boundary from drastically moving toward the minority classes. In detail, we could express the teacher model weight as the following equation:
> > $$
> > \theta^i_t = \hat{\theta} - \gamma \sum_{k=1}^{i-1} (1-\alpha^{-k+(i-1)}) \dfrac{\partial(L_{sup} + \lambda_u L_{unsup}) }{\partial \theta^k_s},
> > $$
> > where $\hat{\theta}$ is the model weight after the burn-in stage, $\theta^i_t$ is the teacher model weight in $i$th iteration, $\theta^k_s$ is the student model weight in $k$th iteration, $\gamma$ is the learning rate, $\alpha$ is the EMA decay (which we use 0.9996),  $L_{sup}$ and $L_{unsup}$ are supervised and unsupervised loss respectively, and ${\lambda}_u$ is the unsupervised loss weight.
> > The regularization of the previous teacher model is equivalent to putting an additional small coefficient on the gradients on student models in previous steps. With the slowly altered decision boundary (*i.e.,* higher stability), the pseudo-labels of these unlabeled instances are less likely to also change dramatically, and the teacher model thus generates more stable pseudo-labels. We have added this discussion in Section 3.3 of the revised paper.
> > - **Better unbiased student:** The teacher model’s predictions in the current step are processed to pseudo-labels and then guide the student model’s prediction in the next step. Thus, if the teacher’s prediction is more stable in its predicted class distribution, the student trained on more balanced pseudo-labels is less likely to be biased to specific classes. This avoids the consecutive biased prediction problem described in the main paper. (Pseudo-labeling methods for semi-supervised tasks aggravate the class-imbalance issue that exists in object detection, as pseudo-labeling methods can be regarded as a closed-loop system.)
> > - To further empirically examine the effectiveness of EMA on imbalance, we present the pseudo-label distribution in different training iterations as presented in the following table (we also present the corresponding figure in the Appendix of the revised paper). At the beginning of training (*i.e.,* 30k), both teacher models with and without EMA could generate the balanced pseudo-labels (the KL divergence between ground-truth labels and pseudo-labels are both small). However, since the Student model is trained with the pseudo-labels generated by the Teacher models, the model without EMA starts biasing towards specific classes. In contrast, with the EMA training, the model generates less imbalanced pseudo-labels. Note that, although the EMA is applied, the balance issue still exists. We thus apply Focal loss to enhance the ability to mitigate the imbalance issue further.
> > |             	| $30K$ iterations | $140K$ iterations |
> > |:-----------:	|:--------------:	|:---------------:	|
> > | Without EMA 	|     $0.0920$     	|      $1.7915$     	|
> > |   With EMA  	|     $0.0730$     	|      $0.2482$     	|
> >
> > We appreciate the reviewer’s constructive feedback, and we have added the above discussions in Section 3.3 and the appendix of the revised paper.

---

> > ### Comment · ~Joy_Hu1 · 2021-06-18
> > **About STAC w Focal Loss**
> >
> > Hi, I have several questions:
> > 1. what is the difference between your method and STAC when wo EMA. Is the difference that your teacher model is continuously updated while STAC's is fixed?
> >
> > 2. How you add focal loss to the model? Did you replace cross-entropy loss with focal loss in both the burn-in stage and the Teacher-Student Mutual Learning stage?
> >
> > As you mentioned in the ablation study, "the model trained with the Focal loss can ... improve ... mAP from 13.42 to 17.85", I think by adding focal loss, the teacher model of your method and STAC should both generate better pseudo-labels. So even STAC fixes the teacher model, it can get some merits from it.

---

### Official Review · AnonReviewer3 · 2020-10-29
**Good paper with solid experiments**

**Rating:** 7
**Confidence:** 4

**Review:**

Paper summary:
This paper focuses on the pseudo-labeling bias issue in semi-supervised object detection (SS-OD), and proposes an Unbiased Teacher framework to address this issue. More specifically, the unbiased teacher framework combines the Mean Teacher model for semi-supervised image classification (Tarvainen and Valpola 2017) and Focal Loss (Lin et al. 2017) for fully supervised object detection to address the bias issue. Experiments on COCO and PASCAL VOC show that the proposed method obtains the state-of-the-art semi-supervised object detection results.


Pros:

+ The paper is well written and easy to understand.

+ The motivations of this paper are clear and interesting.

+ A very simple but effective solution is proposed.

+ Very solid experiments are conducted.


Cons:

- The main weakness of this paper is that the proposed method is a simple combination of the previous methods including Mean Teacher and Focal Loss.
In addition, for the foreground/background imbalance issue, it is straightforward to generate pseudo gts to address this issue, which also has been studied by previous semi-supervised object detection work (Sohn et al. 2020). For the class imbalance issue in object detection, it is also straightforward to use Focal Loss to address this issue.

- One minor thing: It would be better to show results of different detectors / CNN backbones.


Review summary:
In summary, I think this is a good semi-supervised object detection paper because of its simple but effective solution and solid experiments. So I would like to give a weak accept to this paper.


Post-rebuttal comments:
The authors have addressed my concerns in their rebuttals. All reviewers give positive comments to this paper. So I would like to give an accept to this paper.

---

> ### Author Response · Authors · 2020-11-18
> **We added the experiments to address the concerns on simple combination of existing works and generalization to other backbones**
>
> We thank the reviewer for the constructive feedback and questions about the proposed method and its generalization to other backbones.
>
> **1. Straightforward to combine STAC (Sohn et al.) and Focal loss?**
> - The key point of this paper is to show that in order to address semi-supervised object detection, one needs to also address imbalance problems that exist in object detection and exacerbated by the pseudo-labeling process, rather than directly using the state-of-the-art classification techniques on object detection tasks.
> - We regard semi-supervised object detection as a **semi-supervised imbalance issue for classification and regression tasks**, rather than a simple extension task of semi-supervised image classification. We also observed that simply using the SOTA classification technique without considering imbalance issues leads to even worse performance than the supervised-only baseline as shown in Figure 5 (blue curve). On the other hand, as shown in the following table, simply combining STAC (Sohn et al.) and focal loss cannot perform satisfactory results, since the pseudo-labels generated by STAC are not further refined.
> |               	| STAC w/ Cross-entropy 	    | STAC w/ multi-class Focal loss 	    |
> |---------------	|:---------------------:    	    |:------------------------------:	    |
> | $AP^{50:95}$ 	|         $13.97$             	|             $ 13.94$                 	|
>
> - Another merit of our model is simplicity, which enables several extensions to the semi-supervised tasks based on object detectors (*e.g.,* scene graph parsing). We agree that there could be other sophisticated methods that can potentially improve in comparison with Unbiased Teacher, but we would leave this for future research.
> ---
> **2. The generalization to other backbones?**
>
> - To demonstrate the proposed method is generic to other backbones, we apply unbiasedTeacher method on ResNeXt101-FPN [1] and present the COCO-standard 1% setting in the following table. We observe that Unbiased Teacher on ResNeXt101-FPN could also significantly improve against the supervised-only baseline, and ResNeXt101-FPN could achieve similar improvement as ResNet50-FPN did. This confirms the generalization of Unbiased Teacher to different backbones.
> |                	| Supervised-only 	|  Ours  	| Improvement 	|
> |----------------	|:---------------:	|:------:	|:-----------:	|
> |  ResNet50-FPN  	|      $9.05$      	| $20.75$  	|    $11.70$    	|
> | ResNext101-FPN 	|      $13.87$      	|  $25.10$ 	|    $11.23$    	|
>
> [1] “Aggregated Residual Transformations for Deep Neural Networks”, Xie et al., CVPR 2017

---

### Official Review · AnonReviewer4 · 2020-10-29
**Simple yet effective solution**

**Rating:** 9
**Confidence:** 4

**Review:**

This paper addressed an essential task for large-scale application of object detection -- semi-supervised learning. It introduced a simple but effective Unbiased Teacher to solve the traditionally problematic data imbalance issue.

Pros

(1) Provides strong evidences and analysis on the class-imbalance problem inherited in pseudo-labeling methods;
(2) Proposed
      (a) an interesting learning paradigm where the teacher is the temporal ensemble of student networks;
      (b) focal loss in place of cross entropy;
(3) The experiment and ablation suggested that the resulting teacher model is not prone to class-imbalance-induced overfitting and the improvement from SOTA is significant.

Cons

Curious to know why there is a drop in AP_50 comparing to STAC.

---

> ### Author Response · Authors · 2020-11-18
> **We added an explanation on the AP50 metric**
>
>
> We thank the reviewer for the positive feedback and questions about the evaluation metric. We provide the response in the following paragraph.
>
> **1. Why is there a drop in $AP^{50}$?**
> - As mentioned in [1, 2],  $AP^{50}$ is a saturated metric for evaluating object detection tasks, since it counts predicted bounding boxes with IoU larger $0.5$ as correct predictions. This makes all methods perform close to upper-bound (using VOC07+VOC12 labeled set can perform $80.94$%).
> - In the case of VOC07 (labeled) + VOC12 (unlabeled), our number reported in the paper ($77.37$%) only performs $0.08$% lower than STAC ($77.45$%) in the metric of  $AP^{50}$. The reported value is from a single run. We conducted an additional run, and it achieved $77.71$% $AP^{50}$. We note that both methods have similar performance on the metric  $AP^{50}$ (*i.e.,* within the range of variance), while our model could significantly achieve $4.03$% absolute improvement in the metric of  $AP^{50:95}$, which is regarded as a more challenging metric for evaluating object detectors.
>
> [1] “Cascade R-CNN: Delving into High Quality Object Detection”, Cai et al., CVPR 2018\
> [2] “A Simple Semi-Supervised Learning Framework for Object Detection”, Sohn et al., arXiv 2020

---

### Official Review · AnonReviewer2 · 2020-10-29
**effective approach and good results**

**Rating:** 6
**Confidence:** 3

**Review:**

This work tackles the task of semi-supervised object detection via a teacher-student method. The authors introduced a training regime where a teacher and student network, who share the same initial weights pre-trained on labeled data, jointly learns on unsupervised data. They find that label imbalance in the object detection task can lead to inefficient pseudo-label training under the usual SSL training pipeline, and therefore proposes to train the teacher network via exponential moving average to avoid bias in pseudo-labels.


Pros:
1. Revisiting of pseudo-labeling bias issue in semi-supervised object detection is good.
This paper analyze both classification and regression loss with different ratios of labelled data, which shows that classification branch can easily suffer from overfitting that limits current state-of-the-art semi-supervised object detectors. It shows that the misalignment between classification and regression branch not only exists in fully-supervised learning object detection, but also in self-supervised object detection.

2. Experiments are solid and convincing.
Table 1 & Table 2 show that unbiased teacher consistently improves state-of-the-art methods CSD and STAC by a large margin on both COCO and VOC dataset.

Cons:
1. Process of VOC12 dataset.
In Table 2, since VOC12 has a large overlap with VOC07, is the VOC07 part removed from VOC12 as unlabeled data?

2. Novelty
Nevertheless, some contributions seem a bit straight-forward and without significant novelty. The use of EMA is a direct adaptation from successful methods from classification, while Focal loss is already known to tackle class imbalance. The authors might need to provide more insights on the use of these methods (e.g. explaining in more detail how EMA alleviates such bias).

---

> ### Author Response · Authors · 2020-11-18
> **We clarified the concern on datasets and provided an interpretation on how we address pseudo-labeling bias**
>
> We thank the reviewer for raising the dataset question and discussion on the pseudo-labeling bias problem. We have clarified the questions in the following paragraphs.
>
> **1. Is the VOC07 part removed from VOC12 as unlabeled data?**
> - For a fair comparison, we follow the experimental setting of CSD and STAC. Also, we note that there is no overlap between VOC07 and VOC12 sets for __detection__ (we also ran a script to verify there is no duplicated image between these two sets).
> - To the best of our knowledge, the VOC 12 trainval set for __detection__ is a set of images collected from VOC08 to VOC12, which do not overlap with VOC07, while VOC12 trainval for __segmentation__ is a set of images collected from VOC07 to VOC12 and thus has overlap with VOC07.
>
> ---
> **2. The novelty of proposed method**
> - The key point of this paper is to show that in order to address semi-supervised object detection, one needs to also address imbalance problems that exist in object detection and exacerbated by the pseudo-labeling process, rather than directly using the state-of-the-art classification techniques on object detection tasks.
> - We regard semi-supervised object detection as a __semi-supervised imbalance issue for classification and regression tasks__, rather than a simple extension task of semi-supervised image classification. We observed that simply using the SOTA classification technique without considering imbalance issues leads to even worse performance than the supervised-only baseline as shown in Figure 5 (blue curve) of the main paper.
> - Another merit of our model is simplicity, which enables several extensions to the semi-supervised tasks based on object detectors (*e.g.,* scene graph parsing). We agree that there might be other sophisticated methods that can potentially improve the performance in comparison to Unbiased Teacher, but we leave this for future research.

---

> > ### Author Response · Authors · 2020-11-18
> > **Cont’d**
> >
> > **3. How does EMA alleviate biased pseudo-labeling issues?**
> >
> > - The reasons why EMA can alleviate biased pseudo-labeling issues are two-fold:
> >     - In the teacher refinement stage, EMA **improves the stability (*i.e.,* consistent balance) of the teacher model** by preserving the previous teacher model weights.
> >     - In the student learning stage, EMA **makes the prediction of the student model more balanced** by using more stable targets (*e.g.,* pseudo-labels) from a more balanced teacher.
> > - **What if EMA is not used?** When EMA is not used and the teacher model does not preserve the previous teacher model weights, the teacher’s prediction will gradually bias to the majority class. This is because biasing prediction toward the majority classes enables the model to minimize the training loss by easily predicting the majority classes for most instances. Thus, the new decision boundary of the teacher model moves towards the minority classes (*i.e.* pushes into the feature space of the minority class, hence causing the model to classify them as the majority classes). As the pseudo-labeling method uses the previous teacher’s prediction as another supervision for the student in the next step, the pseudo-labeling, which is designed to address low-label scenarios, aggravates the imbalanced prediction issue.
> >
> > - **Higher stability of teacher model:** With the EMA mechanism, the new teacher model is regularized by the previous teacher model, and this prevents the decision boundary from drastically moving toward the minority classes. In detail, we could express the teacher model weight as the following equation.
> > $$
> > \theta^i_t = \hat{\theta} - \gamma \sum_{k=1}^{i-1} (1-\alpha^{-k+(i-1)}) \dfrac{\partial(L_{sup} + \lambda_u L_{unsup}) }{\partial \theta^k_s},
> > $$
> > where $\hat{\theta}$ is the model weight after the burn-in stage, $\theta^i_t$ is the teacher model weight in $i$th iteration, $\theta^k_s$ is the student model weight in $k$th iteration, $\gamma$ is the learning rate, $\alpha$ is the EMA decay (which we use 0.9996),  $L_{sup}$ and $L_{unsup}$ are supervised and unsupervised loss respectively, and ${\lambda}_u$ is the unsupervised loss weight.
> > The regularization of the previous teacher model is equivalent to putting an additional small coefficient on the gradients on student models in previous steps. With the slowly altered decision boundary (*i.e.,* higher stability), the pseudo-labels of these unlabeled instances are less likely to also change dramatically, and the teacher model thus generates more stable pseudo-labels. We have added this discussion in Section 3.3 of the revised paper.
> > - **Better unbiased student:** The teacher model’s predictions in the current step are processed to pseudo-labels and then guide the student model’s prediction in the next step. Thus, if the teacher’s prediction is more stable in its predicted class distribution, the student trained on more balanced pseudo-labels is less likely to be biased to specific classes. This avoids the consecutive biased prediction problem described in the main paper. (Pseudo-labeling methods for semi-supervised tasks aggravate the class-imbalance issue that exists in object detection, as pseudo-labeling methods can be regarded as a closed-loop system.)
> > - To further empirically examine the effectiveness of EMA on imbalance, we present the pseudo-label distribution in different training iterations as presented in the following table (we present the corresponding figure in the Appendix of the revised paper). At the beginning of training (*i.e.,* 30k), both teacher models with and without EMA could generate the balanced pseudo-labels (the KL divergence between ground-truth labels and pseudo-labels are both small). However, since the Student model is trained with the pseudo-labels generated by the Teacher models, the model without EMA starts biasing towards specific classes. In contrast, with the EMA training, the model generates less imbalanced pseudo-labels. Note that, although the EMA is applied, the balance issue still exists. We thus apply Focal loss to enhance the ability to mitigate the imbalance issue further.
> > |             	|$30K$ iterations   |   $140K$ iterations  |
> > |:-----------:	|:--------------:	|:---------------:	|
> > | Without EMA 	|     $0.0920$     	|      $1.7915$     	|
> > |   With EMA  	|     $0.0730$     	|      $0.2482$     	|
> >
> > We appreciate the reviewer’s constructive feedback, and we have added the above discussions in Section 3.3 and the appendix of the revised paper.

---

### Decision · Program_Chairs · 2021-01-07
**Final Decision**

**Decision:**

Accept (Poster)

**Comment:**

This paper proposed a new semi-supervised object detection approach using Unbiased Teacher to jointly address the pseudo-labeling bias and overfitting issues. Significant improvements over SOTA were reported on COCO and VOC. Reviewers agree that the proposed method is simple and effective, and the experimental results are solid and convincing.  While the novelty of technical contributions for individual components may not be very significant, the idea is simple and well executed with strong results and good presentation. Overall, the paper is recommended for acceptance (poster).